# Bayesian Optimization with Support Vector Machine Model for Parkinson Disease Classification

**DOI:** 10.3390/s23042085

**Published:** 2023-02-13

**Authors:** Ahmed M. Elshewey, Mahmoud Y. Shams, Nora El-Rashidy, Abdelghafar M. Elhady, Samaa M. Shohieb, Zahraa Tarek

**Affiliations:** 1Computer Science Department, Faculty of Computers and Information, Suez University, Suez 43512, Egypt; 2Faculty of Artificial Intelligence, Kafrelsheikh University, Kafrelsheikh 33516, Egypt; 3Deanship of Scientific Research, Umm Al-Qura University, Makkah 21955, Saudi Arabia; 4Information Systems Department, Faculty of Computers and Information, Mansoura University, Mansoura 35561, Egypt; 5Computer Science Department, Faculty of Computers and Information, Mansoura University, Mansoura 35561, Egypt

**Keywords:** Parkinson’s disease, Bayesian Optimization, support vector machine, hyperparameter tuning, classification, evaluation metrics

## Abstract

Parkinson’s disease (PD) has become widespread these days all over the world. PD affects the nervous system of the human and also affects a lot of human body parts that are connected via nerves. In order to make a classification for people who suffer from PD and who do not suffer from the disease, an advanced model called Bayesian Optimization-Support Vector Machine (BO-SVM) is presented in this paper for making the classification process. Bayesian Optimization (BO) is a hyperparameter tuning technique for optimizing the hyperparameters of machine learning models in order to obtain better accuracy. In this paper, BO is used to optimize the hyperparameters for six machine learning models, namely, Support Vector Machine (SVM), Random Forest (RF), Logistic Regression (LR), Naive Bayes (NB), Ridge Classifier (RC), and Decision Tree (DT). The dataset used in this study consists of 23 features and 195 instances. The class label of the target feature is 1 and 0, where 1 refers to the person suffering from PD and 0 refers to the person who does not suffer from PD. Four evaluation metrics, namely, accuracy, F1-score, recall, and precision were computed to evaluate the performance of the classification models used in this paper. The performance of the six machine learning models was tested on the dataset before and after the process of hyperparameter tuning. The experimental results demonstrated that the SVM model achieved the best results when compared with other machine learning models before and after the process of hyperparameter tuning, with an accuracy of 92.3% obtained using BO.

## 1. Introduction

Parkinson’s disease (PD) is a recognized clinical illness with a variety of etiologies and clinical manifestations. According to current definitions, PD is defined as the presence of bradykinesia together with either rest tremor, stiffness, or both. In the majority of populations, genetic factors connected to known PD genes account for 3–5% of PD, which is referred to as monogenic PD. In contrast, 90 genetic risk variations account for 16–36% of the heritable risk of non-monogenic PD. Constipation, being a non-smoker, having a relative with PD or tremor, and the additional causative factors all at least double the chance of PD. There is currently no treatment that can slow or stop the course of PD, however new knowledge about its genetic origins and processes of neuronal death is being developed [1].

### 1.1. Problem Statement

The use of machine learning (ML) algorithms is becoming increasingly common in the medical industry. As its name indicates, ML enables software to train data and develop outstanding representations in a semi-automatic manner. For the purpose of diagnosing Parkinson’s disease (PD), several data formats have been applied to ML approaches. ML also makes it possible to combine data from many imaging systems in order to identify Parkinson’s disease. In order to rely on these different measures for diagnosing Parkinson’s disease in preclinical phases or atypical structures, relevant characteristics that are not typically utilized in the diagnosing of Parkinson’s disease are discovered through the application of ML algorithms. This allows for the diagnosis of Parkinson’s disease in earlier stages. In recent years, there has been an increase in the number of publications published that discuss the use of ML to diagnose PD. Earlier studies did investigate the use of ML in the diagnosis and assessment of Parkinson’s disease, but they were only able to evaluate inputs from sensing devices and motor and kinematics symptoms [2]. Computer-based statistical methods known as machine learning algorithms may be trained to look for recurring patterns in large volumes of data. Clinicians can use machine learning techniques to identify patients based on several criteria at once [3]. 

### 1.2. Objectives

It is possible to use model-based and model-free strategies to predict certain medical outcomes or diagnostic characteristics. Generalized linear models are an illustration of model-based techniques. One of the most often used model-based techniques is logistic regression, which is useful when the output parameters are assessed on a binary scale (e.g., failure/success) and follow the distribution of Bernoulli. Therefore, using the predicted probabilities as a basis, categorization may be performed. The model assumptions must be thoroughly examined, verified, and the right connection functions must be chosen by the investigators. Because the statistical principles may not always apply in real-world circumstances, particularly for significant volumes of incongruent data, the model-based procedures may not be applicable or may provide biased conclusions. This is especially the case if there are massive quantities of incongruent data. Model-free approaches, on the other hand, make less assumptions and accommodate the underlying characteristics of the data without having to build any models in advance. Model-free approaches, such as Random Forest, Support Vector Machines, AdaBoost, Neural Networks, XGBoost, and SuperLearner are capable of building non-parametric interpretations, which are also known as (non-parametric) techniques, from difficult data without simplifying the issue. Since these algorithms do not provide ideal classification/regression outcomes, they benefit from ongoing learning or retraining. Nevertheless, model-free ML algorithms offer significant promise for tackling real-world issues when properly maintained, as well as trained and reinforced effectively [4]. The accurate and early identification of PD is critical because it can reveal valuable information that can be used to slow down the course of the disease [5].

Classification has a purpose in PD identification to reduce time and improve treatment effectiveness. The challenge is to find the classification method that is most effective for PD detection; however, a study of the relevant knowledge reveals that various different classification techniques have been employed to provide superior outcomes. The difficulty in choosing the best classification method is that it must be applied to a local dataset. 

### 1.3. Paper Contribution 

In this study, Bayesian Optimization is used to optimize the hyperparameters for six machine learning models, namely, Random Forest (RF), Support Vector Machine (SVM), Naive Bayes (NB), Logistic Regression (LR), Ridge Classifier (RC), and Decision Tree (DT) to determine the categorization method that is both the most effective and precise for PD. The dataset used consists of 23 features and 195 instances. The experimental results demonstrated that the SVM model achieved the best outcomes when compared with various ML models before and after the process of hyperparameter tuning, with an accuracy 92.3% obtained using BO. 

### 1.4. Paper Organization 

The remaining sections of paper are arranged in the following order. Section 2 describes a comprehensively summary of some studies published that used machine learning techniques in the diagnosis and classification of PD to provide a comprehensive overview of data source, sample size, ML techniques, associated outcomes, and benefits and limitations. Section 3 presents the proposed BO for ML models in PD categorization. Section 4 shows the evaluation of the proposed approach and comparison with various ML approaches in the classification of Parkinson’s disease. Section 5 discusses the conclusions of this study.

## 2. Related Work 

Parkinson’s disease is considered the second neurodegenerative condition, which is characterized by low dopamine levels in the brain, following Alzheimer’s disease (AD) [1,6]. The early diagnosis of PD contributes to saving and improving a patient’s quality of life [7]. However, most of the literature emphasized that identifying PD at an early stage considers a challenge. PD has mainly characterized by four symptoms, including postural instability, rigidity, bradykinesia, and tremor [8]. PD diagnosis is traditionally based on motor symptoms, as it is the most obvious symptom and most of the rating scales utilized them for PD evaluation. Despite the importance of the non-motor symptoms (i.e., olfactory dysfunction, sleep disorder, voice change) in early prediction, their complexity yields variability among patients. Therefore, these symptoms are not used for PD diagnosis [9]. In the last decades, several studies have utilized machine learning and deep learning in several diagnosis and prediction problems in the healthcare domain [10,11]. For PD diagnosis and prediction, ML has been applied to different data modalities, including movement [12], handwriting [13], patients’ neuro images [14,15], cardiac scintigraphy [16], magnetic resonance imaging (MRI) [17], and optical coherence tomography (OCT) [18]. 

Das [19] utilized several ML models based on movement data with a neural network classifier (NN) and obtained promising results according to several evaluation metrics. Åström and Koker [20] utilized parallel NN to improve the classification performance. The developed model was evaluated using a rule-based system for the parallel NN to improve the total performance by about 8.2%. Another study presented by Bhattacharya and Bhatia [21] applied support vector machine (SVM) with different kernels after applying data preprocessing, evaluating results with the receiver operation curve (ROC). Chen et al. [22] provided a diagnosis model based on a fuzzy k-nearest neighbor model. They compared the performance of FKNN with SVM after applying a principal component analysis for data compression. Their proposed model achieved 92% in terms of classification accuracy. ML also allows studies to combine more than data modalities to make a prediction. For example, Li et al. [23] provide a fuzzy-based system to make a nonlinear transformation, then the transformed data is reduced to another dimension based on PCA, then SVM is utilized for PD prediction. Eskidere et al. [24], compare the performance of SVM, least square SVM, and Multilayer perceptron. The results show that the least square SVM gives the best performance among all classifiers. Nilashi et al. [25] took advantage of clustering and classification for PD prediction. They first used EM and PCA for multi-collinearity then applied a neuro-fuzzy inference system and SVR. Guo et al. improved the performance by developing learning feature functions based on a genetic algorithm (GA) and expectation maximization (EM). 

Peterek et al. [26] examined the performance of random forest (RF) for the diagnosis and tracking of PD disease. With the advancement in PD medical diagnosis, several studies [27,28] affirmed that PD symptoms vary in degree and combination among PD patients. However, more than 90% of PD patients suffer from vocal impairment. Therefore, several recent studies in PD diagnosis contexts pay attention to patients’ voices and the change in their phenotypes. Consequently, this is utilized as an early symptom in PD prediction in various studies as shown in Table 1. In [29], the accuracy shows that the leave-one-out cross validation (LOOCV) was 63.20% when PD patients were compared to healthy control patients, with a particular number of patients gathered and analyzed using the C4.5 decision tree (DT). In [30], a dataset of 42 participants from Parkinson’s disease patients is obtained using the Least Absolute Shrinkage and Selection Operator (LASSO) and the smallest absolute error is 8.38. Shahid and Singh [31] proposes a deep learning strategy for ten Parkinson’s disease patients with a determination coefficient (R2) of 0.956. Fernandes et al. [32] provided a dataset of wearable sensors positioned on both feet from 15 IPD, 15 VaP, and 15 healthy participants. The accuracy of multi-layer perceptron and deep belief neural networks was 94.50% and 93.50%, respectively. The authors in [33] presented an algorithm based on SVM for 13 PD patients and they achieved 83.00% accuracy. The fuzzy system and neural networks are combined to diagnose PD, as presented by [34]. They reach an accuracy of 81.03 percent using two hidden layers and a small sample of 31 people, 23 of whom are classified as having Parkinson’s disease. A multi-task learning approach for predicting Parkinson’s disease using medical imaging data was presented by [35] and the model achieved an accuracy between 80.00% and 92.00%. A feature selection approach based on iterative canonical correlation analysis (ICCA) was used to study the involvement of various brain areas in PD using T1-weighted MR images presents by [36]. Drotár et al. [37] leveraged the fact that movement during text handwriting comprises not only on-surface hand motions, but also in-air trajectories conducted as the hand moves in the air between one stroke to the next. In 37 PD patients on medication and 38 age- and gender-matched healthy controls, they utilized a digitizing tablet to analyze both in-air and on-surface kinematic factors while scribbling a text.

## 3. Materials and Methods

These days, Parkinson’s disease (PD) is very prevalent all over the world. The human nervous system and numerous bodily components that are linked by nerves are impacted by PD. The ML models development that can aid in disease prediction can be extremely important for early prediction. In this study, we use a common dataset and a few machine learning techniques to classify Parkinson’s patients. Before assessing a performance of a model, hyper-parameter optimization enables fine-tuning. The Bayesian Optimization (BO) approach is utilized to generate samples of hyper-parameter values in order to discover the optimum values. Classification approaches are trained using a training set for optimization and tested using a test set for each hyper-parameter configuration. The ideal parameter setup is the one that provides the highest overall accuracy. The following phase involves training each model using the original training set’s optimums, and the accuracy is assessed by classifiers on the test set. In this study, the hyperparameters for six machine learning models, namely, SVM, RF, LR, NB, RC, and DT, are optimized using Bayesian optimization (BO). Twenty-three features and 195 instances make up the dataset used in this study. Accuracy, recall, F1-score, and precision were computed as evaluation measures to assess the effectiveness of the supposed categorization models. Using the dataset, the six machine learning models’ performance was evaluated both before and after the hyperparameter tuning procedure, and the experiments showed that support vector machine is the optimal classifier among the utilized classifiers. Figure 1 shows the proposed Bayesian Optimization for various machine learning (ML) models in Parkinson’s disease categorization.

In this research, we use a real-world dataset to develop a hybrid BO-SVM model for classifying patients with Parkinson’s disease. Separate portions of the entire dataset are used for training and testing purposes. Models of classifiers may be built using training data. Later, the created models are scored by how well they categorize the test data. In order to construct an effective classification model for Parkinson’s disease, SVM is supposed with Bayesian Optimization for tuning hyperparameters. Identifying the variables you want to use as predictors and the result you want to obtain are the initial steps in developing a classification model of SVM. The next step is to run searches to fine-tune the SVM’s hyperparameters.

Finally, the tuned hyperparameters of SVM are used in classification, and the model’s performance is evaluated using test data. According to experimental results, SVM achieved 89.6% before hyperparameter tuning compared to 80.9%, 82.1%, 85.7%, 85.3%, and 87.2% for RC, NB, DT, LR, and RF, respectively. After applying hyperparameter tuning, there were two hyperparameters of SVM: Kernel = rbf, and regularization parameter (C) = 0.4. The SVM with BO achieved 92.3% compared to 83.3%, 84.6%, 88.5%, 87.2%, and 89.7% for BO with NB, DT, LR, and RF, respectively. Therefore, it is worth utilizing the SVM method for Parkinson’s disease classification in conjunction with other ML algorithms. 

### 3.1. Min-Max Normalization 

A crucial step in any analysis that compares data from multiple domains is normalization. Normalization moves information from a given domain to a range, such as between (0, 1). Numerous techniques exist for normalizing data, such as decimal scaling, min-max, Z-score, median-mad, mean-mad, and norm normalization techniques [38]. The min-max normalization approach rescales the property from its domain to a new set of values, such as between (0, 1). The basis of this approach is as follows:(1)$$f\left(n\right)=\frac{n-\min\left(n\right)}{\max\left(n\right)-\min\left(n\right)}$$
where f(n) is the normalized features, and n is the input feature value. The max(n) and min(n) are the highest and lowest sets of the input feature.

### 3.2. Bayesian Optimization

Hyperparameters are a group of factors used in testing and training to support the learning process. The learning rate, iterations number, batch size, hidden layers, momentum, regularization, and activation functions are examples of hyperparameters. The parameters might be an integer or categorical or continuous variable with values ranging from the lower to higher bounds. Hyperparameters are stable throughout the training process, which improves model accuracy while simultaneously reducing memory usage and training time. Based on the problem description, different models use different hyperparameters. There is no optimum hyperparameters that apply to all models [39]. 

The term “Bayesian Optimization (BO)” refers to a method that may be used in a sequential fashion to optimize the parameters of any black-box function f(x). BO integrates prior belief for the purpose of evaluating a response surface function fˆ(x), utilizing fˆ(x) to choose the configuration xn to try, evaluates f(xn) by using true f(x), specifies posterior belief through assessed performance f(xn), and continues the procedure in sequential manner until a stop criteria is arrived at to tune the test sample for achieving enhanced parameters that collaborate for better classification [40]. The framework of BO is shown in Figure 2.

The Bayesian theorem forms the basis of BO [41,42]. In order to update the optimization function posterior, it establishes a prior over the optimization function and collects information from the previous sample set [43]. Equation (2), which asserts for a model A and observation B, is the foundation for the optimization process that based on Bayes’ Theorem [44].
(2)$$P\left(A|B\right)=(P(B|A)P\left(A\right))/P\left(B\right)$$
where P(A|B) denotes the likelihood of A given B, P(B|A) represents the likelihood of B given A, P(A) indicates the prior probability of A, and P(B) signifies the marginal probability of B. Bayesian Optimization is utilized to determine the minimal value of a function on a limited set [39]. 

### 3.3. Machine Learning Models Using Bayesian Hyperparameter Optimization

We present some ML models for Parkinson’s disease categorization. The Bayesian Optimization approach is used to fine-tune hyperparameters for six popular ML models: SVM, RF, LR, NB, RK, and DT. The Machine Learning Repository (UCI) dataset [45] was used to assess the classifiers’ efficiency. BO is a hyperparameter tuning method for improving the accuracy of machine learning models. BO seeks to collect observations that disclose as much information as possible about the function and the position of its optimal value With Bayesian Optimization, the ideal value might be discovered using relatively few samples. It does not need the explicit formulation of the function, in contrast to conventional optimization techniques. Therefore, Bayesian Optimization is ideal for hyperparameter tuning. Therefore, initially, BO is applied to tune hyperparameters for the Support Vector Machine (SVM) algorithm [46], Random Forest (RF) [47], Logistic Regression (LR) [48], Naive Bayes (NB) [49], Ridge Classifier (RC) [50], and Decision Tree (DT) [51]. 

SVM is a popular supervised machine learning technique used for both classification and regression tasks; it is based on the kernel method [52]. Because of this, we set out to optimize the SVM hyperparameters in search of the kernel function and parameters that would provide the most reliable model [53,54]. Using a random starting point in the hyperparameter space, the Bayesian technique iteratively assesses prospective hyperparameter configurations in light of the existing model to see if any of them enhance the model. Based on the experimental results presented in this work, the suggested Bayesian Optimization-Support Vector Machine (BO-SVM) achieves the greatest accuracy for the classification process. The pseudocode of proposed approach is presented in Algorithm 1.
**Algorithm 1.** Bayesian Optimization-Support Vector Machine (BO-SVM)**Input**: Dataset D, hyper-parameter space Θ, Target score function T(θ), max n° of evaluation n_max_.**Split** randomly the D into N folds; one for train set and the other for test set. **Build** a model m on the train dataset using SVM approach. **Choose** a starting configuration θ_0_ ϵ Θ. **Assess** the original score y_0_ = T(θ_0_). **Initialize** S_0_ = {θ_0_, y_0_}**While** t < maximum number of iterations do
**For** m = 1, …, m_max_ do
**Choose** a new hyperparameter arrangement θ_m_
ϵ Θ by enhancing function U_m_Θ_m_ = argθ ϵ Θ  max U_m_ (θ, S_t_),**Analyze** H in θ_m_ to get a new numerical score y_m_ = T(θ_m_).**Strengthen** the data Sm=Sm−1 ∪ {θ_m_, y_m_}.**Update** the surrogate model. m = m + 1**End for****End while**
**Extract** optimized hyperparameters. **Build** SVM model using these tuned hyperparameters from the test data set. **Solve** the optimization problem, evaluate the accuracy and save it in array. **Output**: Mean accuracy of classification.

## 4. Experimental Results

### 4.1. Dataset Description

The dataset used in this paper is available at [45]. The dataset consists of 23 features and 195 instances. The data were first created in a collaboration between Oxford University and the National Centre for Voice and Speech by Max Little. They include 195 sustained vowels aggregated from 31 females and males, 23 of them diagnosed with PD. All patients ranged from 46 to 85 (65.8 ± 9.8). The duration from diagnosis was 0 to 28 years. For each subject, a range of biomedical phonetics was recorded which ranged from one to 36 s. The data were recorded using IAC sound with an AKG C420 Microphone that was positioned about 8 cm from the patient’s lips. Then, the voice signals were transferred directly to a computer based on CSL 4300B kay. All voice signals were sampled at 44.2 kHz with 16-bit resolution. Despite amplitude normalization, which affects the calibration, the study mainly focused on changes in the absolute change pressure level. The data were in the ASCII CSV format. Each column represents a specific voice measure, while each row represents one recording from patients. For each patient, there are roughly six recordings with different specific voice measures. The first column in the dataset refers to the patient’s name. Table 2 details the dataset features information. The statistical analysis for the dataset is illustrated in Table 3.

The heatmap analysis for the dataset features is shown in Figure 3. Heatmap analysis is a commonly used technique in data analysis to visualize the relationship between variables in a dataset. It is used to identify strong and weak relationships between features, and to understand how features are correlated with one another. In this figure, the vertical and horizontal bars are the numerical data of the applied features. The numerical data in the heat map are normalized from 0 to 1, the brightness indicates that the value is 1 and the dark color indicates that the value is 0. The diagonal values are 1, which means that the features are totally corelated and when the values decreased, it means the correlation between features is decreased. This statistically helps us to diagnose and prognose the PD in terms of the heatmap figure. Figure 4 shows the Box plot visualization per category label analysis for the dataset characteristics.

Figure 5 demonstrates a box plot for distribution analysis of the features. It is a useful tool for visualizing the distribution of numerical data. When analyzing the distribution of features in a dataset, a box plot can be used to display the distribution of each feature. This type of visualization is called a box plot for distribution analysis of features. In this figure, we visualize the enrolled features which are the most significant 23 features of the applied PD dataset. Box plots split data into portions that each include around 25% of the data in that set. Box plots are valuable because they give a visual overview of the data, allowing us to easily determine mean values, dataset dispersion, and skewness. 

Figure 6 demonstrates the histogram for distribution analysis of the features, which is a graphical representation of the distribution of a dataset, showing the frequency of data points within different intervals. It is a useful tool for visualizing the distribution of numerical data. We explored the histogram of the characteristics in this figure, which is a standard graphing tool used to incorporate discrete and continuous data recorded on an interval scale. It is frequently used to depict the key aspects of data distribution in a convenient format.

### 4.2. Evaluation Metrics 

The experimental results were executed using jupyter notebook version (6.4.6). Jupyter Notebook is a popular tool for data analysis and visualization in Python. It allows you to write and run code, display visualizations, and document your findings all in one place. It runs on a web browser and supports many programming languages, including Python 3.8. The experiment was run using a computer with an Intel Core i5 processor and 16 GB RAM, using the Microsoft Windows 10 operating system. In this paper Bayesian Optimization is used to optimize the hyperparameters for six machine learning classification models, namely, Support Vector Machine (SVM), Random Forest (RF), Logistic Regression (LR), Naive Bayes (NB), Ridge Classifier (RC), and Decision Tree (DT). The performance of BO-SVM model was compared with several machine learning models. The performance of the classification models utilized in this article was measured using four different metrics: accuracy, recall, precision and F1 score. Accuracy is calculated using Equation (3):(3)$$\mathrm{Accuracy}=\frac{TP+TN}{TP+FP+FN+TN}$$
where TP if true positive, TN is true negative, FP is false positive, and FN is false negative. Recall is calculated using Equation (4):(4)$$\mathrm{Recall}=\frac{TP}{TP+FN}$$

Precision is calculated using Equation (5):(5)$$\mathrm{Precision}=\frac{TP}{TP+FP}$$

F1 score is computed using Equation (6):(6)$$F1\;\mathrm{score}=\frac{2*Recall*Precision}{Recall+Precision}$$

The hyperparameters for the classification models in the experimental were optimized using a Bayesian Optimization approach. The best hyperparameters for each model are listed in the Table 4, where:Random Forest (RF): The best number of estimators was 10, using the “gini” criterion.Ridge Classifier (RC): The best alpha was 0.4, with “copy_X” set to false, “fit_intercept” set to true, “normalize” set to false, and using the “lsqr” solver with a tolerance of 0.01.Decision Tree (DT): The best criterion was “entropy” and the best splitter was “random”.Naive Bayes (NB): The best alpha was 0.1 and the best value for “var_smoothing” was 0.00001.Logistic Regression (LR): The best penalty was “l2” and the best solver was “lbfgs”.Support Vector Machine (SVM): The best kernel was “rbf” and the best value for the regularization parameter (C) was 0.4.

Table 5 show the performance of each model using the Bayesian Optimization approach in terms of accuracy, F1 score, recall, and precision. The model with the highest accuracy, F1 score, recall, and precision is BO-SVM, with an accuracy of 92.3%, F1 score of 92.1%, recall of 92.3%, and precision of 92.1%. The lowest results among the models are seen in BO-RC, with an accuracy of 83.3%, F1 score of 82.2%, recall of 83.3%, and precision of 82%. Figure 7 represents the accuracy of the classification models using the Bayesian Optimization approach.

Table 6 shows the performance of the classification models in terms of accuracy, using default parameters. The use of default values can simplify the modeling process, as it eliminates the need for manual tuning of hyperparameters. The default values specified by the scikit-learn library are chosen based on general best practices and have been found to work well in a variety of situations. From the results in Table 6, it can be seen that the SVM model has the highest accuracy among the models, with 89.6%. The Random Forest model comes in second with an accuracy of 87.2%. On the other hand, the Ridge Classifier model has the lowest accuracy among the models with 80.9%. It is important to note that these results are based on the default parameters of each model and may be improved through hyperparameter tuning, as demonstrated in Table 5.

It can be concluded that hyperparameter tuning through Bayesian Optimization significantly improves the performance of the models compared to their default parameters. The Bayesian Optimization approach helps to optimize the hyperparameters and results in better accuracy for each of the models. Figure 8 represents the accuracy of the classification models using the default hyperparameters.

The results of the study were further evaluated using confusion matrices, which were presented in Figure 9. These matrices helped to more effectively evaluate the performance of each classifier. The results indicated that the BO-SVM had the best performance, outperforming the other classifiers.

### 4.3. Discussion 

This paper proposes a novel method for distinguishing between those who have Parkinson’s disease (PD) and those who do not, based on Bayesian Optimization-Support Vector Machine (BO-SVM). Bayesian Optimization (BO) with a hyperparameter tuning technique is used to optimize the hyperparameters for six distinct machine learning models, namely, Support Vector Machine (SVM), Random Forest (RF), Logistic Regression (LR), Naive Bayes (NB), Ridge Classifier (RC), and Decision Tree (DT). The dataset utilized in this study has 23 characteristics and 195 occurrences, and the models’ performance was measured using four metrics: accuracy, F1-score, recall, and precision.

The findings revealed that the SVM model performed the best among all models, both before and after hyperparameter tuning, with an accuracy of 92.3 percent reached using BO. The paper presented an essential contribution to the subject of machine learning and its applications in healthcare. For diagnosing speech deficits in patients at the early stages of central nervous system illnesses, Lauraitis et al. [55] used a Bidirectional Long Short-Term Memory (BiLSTM) neural network and a Wavelet Scattering Transform with Support Vector Machine (WST-SVM) classifier (CNSD). The study included 339 voice samples obtained from 15 participants: 7 with early stage CNSD (3 Huntington, 1 Parkinson, 1 cerebral palsy, 1 post stroke, 1 early dementia), and 8 healthy subjects. Their speech data are collected using a voice recorder from the Neural Impairment Test Suite (NITS) mobile application. Features are extracted from pitch contours, mel-frequency cepstral coefficients (MFCC), gammatone cepstral coefficients (GTCC), Gabor (analytic Morlet) wavelets, and auditory spectrograms. Ultimately, 94.50% (BiLSTM) and 96.3% (WST-SVM) accuracy is achieved for solving the healthy vs. impaired classification problem. The developed method can be applied for automated CNSD patient health state monitoring and clinical decision support systems, and as a part of the Internet of Medical Things (IoMT). In this work, we utilized BO with SVM. Therefore, the questions here are: although there are several hyperparameter optimization (HPO) tools, why the choice of BO? Does BO carry any distinct advantages when compared with other HPO methods. Will the ML algorithms give better results when optimized using other methods? In answer to these questions, BO has several advantages compared to other hyperparameter optimization (HPO) methods.

Model-based approach: BO uses a probabilistic model to represent the relationship between the hyperparameters and the performance of the model. This allows BO to make informed decisions about which hyperparameters to try next based on the results of previous trials.Handling of noisy objectives: BO can handle noisy or stochastic objective functions, such as those that may be encountered in real-world machine learning applications.Incorporation of prior knowledge: BO allows for the incorporation of prior knowledge about the objective function through the use of a prior distribution over the hyperparameters.Efficient exploration–exploitation trade-off: BO balances exploration (trying new, potentially better hyperparameters) and exploitation (using the current best hyperparameters) in an efficient manner, allowing for faster convergence to the optimal hyperparameters.

A single observation from the original dataset is utilized as the validation set, also known as the test set, in leave-one-out cross validation (LOOCV), and the remaining observations constitute the training set. This technique is performed N times, with each observation serving as a validation set once. The LOOCV approach was used to measure classifier performance on unseen instances in separate and pooled datasets. The proportion of correct classifications over the N repetitions is used to define performance here. To ensure that the training set’s attributes, and thus the trained classifier’s, are not influenced by the validation sample, the test subject was removed from the initial dataset before applying the training set (with N1 samples), in order to obtain the subject scores required to train the classifier. The classifier was then utilized to determine the test subject’s label [29]. In this work, we do not need to use LOOCV cross validation because we utilized BO optimization, and the achieved results are promising compared with other results, as shown in Algorithm 1 and Figure 2.

We conducted a comparison study using the same standard dataset published in the UCI repository in [45] to compare the proposed model with the latest technique. Li et al. [56] showed NB, 3-NN, SVM-linear, and SVM-poly, with respective accuracies of 66.31%, 67.73%, 53.91%, and 55.41%. Sajal et al. [57] provided a method based on KNN, SVM, and NB, with accuracies of 90.50%, 87.00%, and 81.00% for five levels of classification in tremor analysis. Furthermore, Haritha et al. [58] obtained 76.20%, 86.71%, 91.83%, 82.90%, and 87.03% accuracy utilizing NB, DT, RF, MLP, and LR, respectively. Abayomi-Alli et al. [59] demonstrated a Bidirectional Long Short-Term Memory (BiLSTM) for the UCI PD dataset, and their model achieved an accuracy of 82.86% with the original data. Fang and Liang [60] presented the UCI dataset for Parkinson’s disease and optimization algorithms such as Particle Swarm Optimization (PSO), Whale Optimization Algorithm (WOA), Grasshopper Optimization Algorithm (GOA), Binary PSO (BPSO), and Binary GOA (BGOA) compared to the Nonlinear Binary Grasshopper Whale Optimization Algorithm (NL-BGWOA), and the results showed that the NL-BGWOA achieved 91.30% higher than other optimization algorithms. Figure 10 demonstrates the comparative study of the proposed method based on BO-SVM with the mentioned methods based on the same applied PD standard dataset.

## 5. Conclusions and Future Work

A Bayesian Optimization-Support Vector Machine (BO-SVM) model was proposed for classifying Parkinson’s disease (PD) patients and non-patients in this study. The dataset used consisted of 195 instances with 23 features and the target feature was binary, with 1 indicating PD and 0 indicating no PD. Six machine learning models (SVM, Random Forest, Logistic Regression, Naive Bayes, Ridge Classifier, and Decision Tree) were evaluated using four metrics (accuracy, F1-score, recall, and precision) both before and after hyperparameter tuning using BO. The results showed that SVM outperformed the other models, achieving an accuracy of 92.3% after BO tuning. Future work for this study could include expanding the dataset used to classify Parkinson’s disease to include more diverse and representative sample populations. Additionally, incorporating more advanced machine learning techniques, such as deep learning, could lead to even better results in terms of accuracy and performance. Another area for improvement could be exploring different types of feature selection methods to identify the most important features for the classification task. Finally, validating the results on a separate independent dataset could provide further confidence in the robustness and generalizability of the proposed BO-SVM model. The future direction of this study could be generalization to a larger population and hence potential integration into a larger healthcare system using the Internet of Medical Things and fog computing.

## Figures and Tables

**Figure 1 sensors-23-02085-f001:**
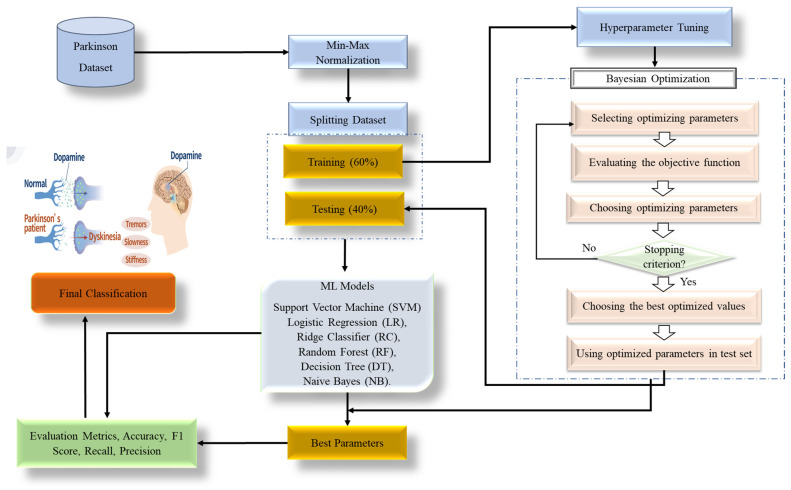
The proposed BO-ML models for Parkinson’s disease classification.

**Figure 2 sensors-23-02085-f002:**
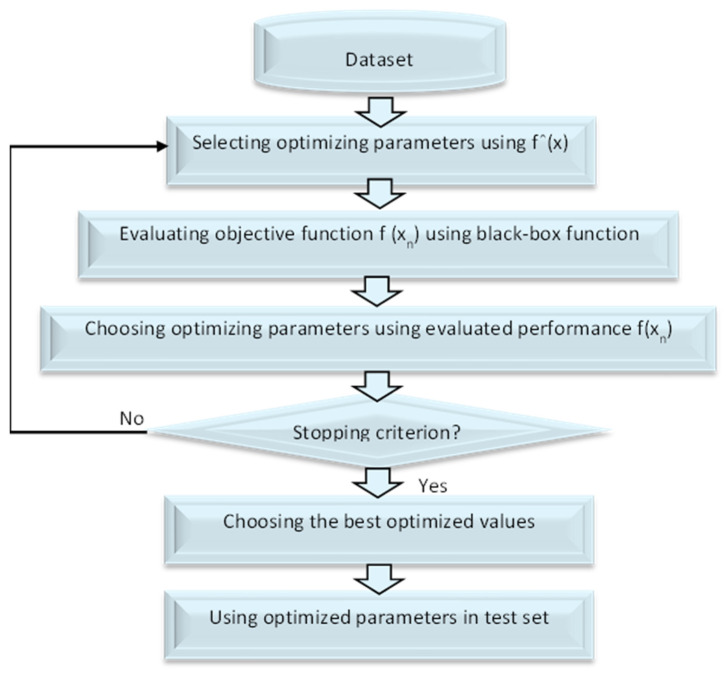
The framework of BO.

**Figure 3 sensors-23-02085-f003:**
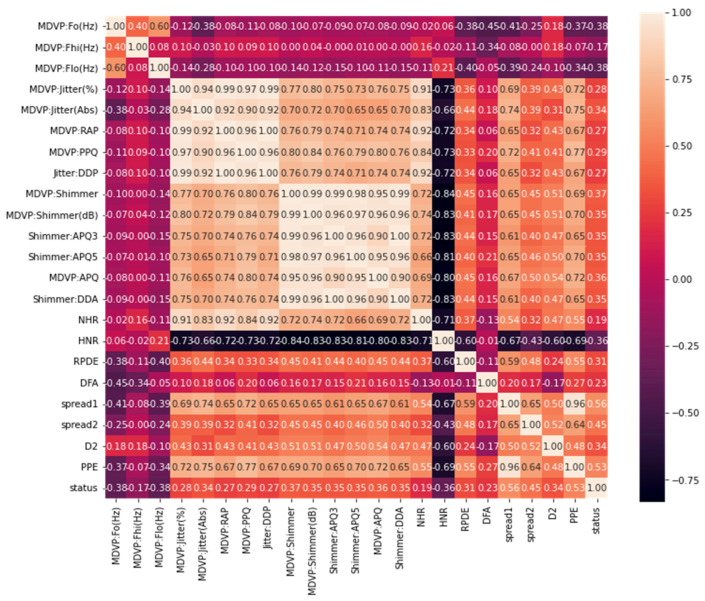
Heatmap analysis for the dataset features.

**Figure 4 sensors-23-02085-f004:**
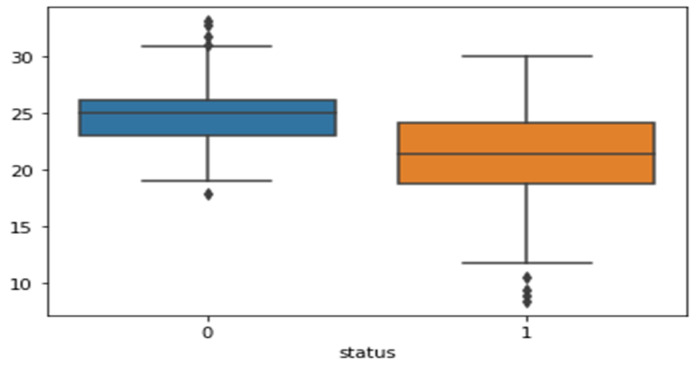
Box plot visualization per category label analysis for the dataset features.

**Figure 5 sensors-23-02085-f005:**
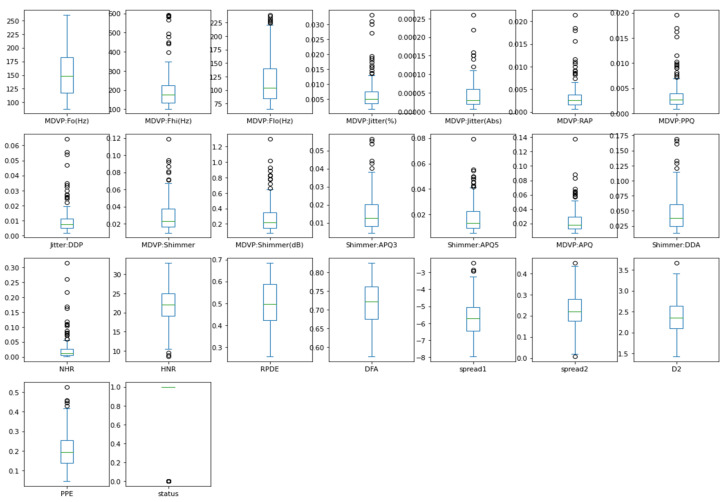
Box plot for distribution analysis of the features.

**Figure 6 sensors-23-02085-f006:**
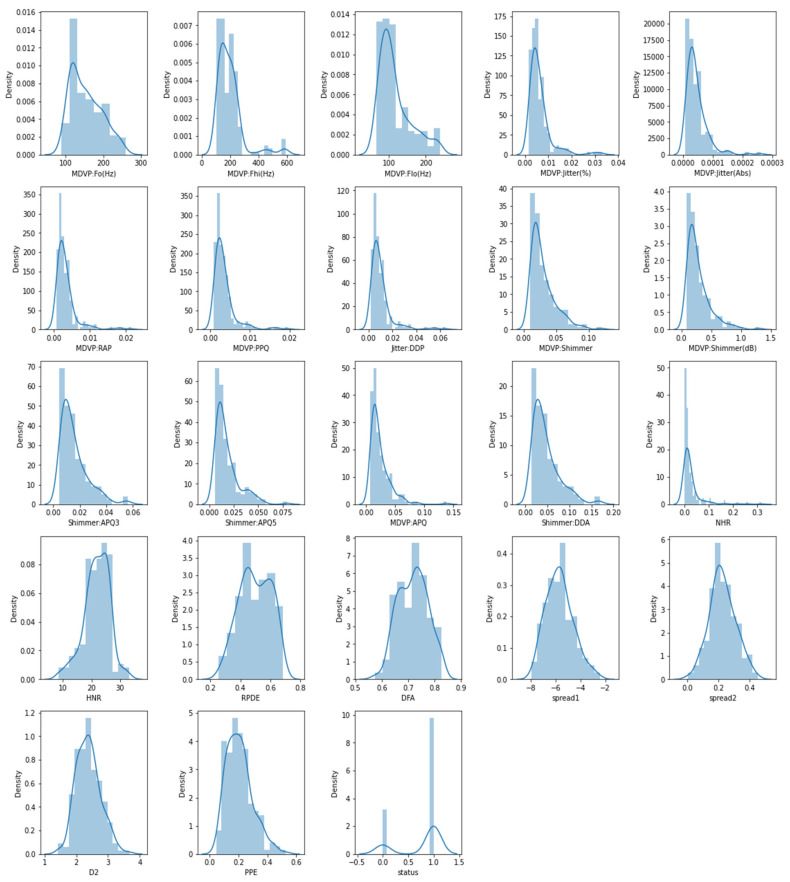
Histogram for distribution analysis of the features.

**Figure 7 sensors-23-02085-f007:**
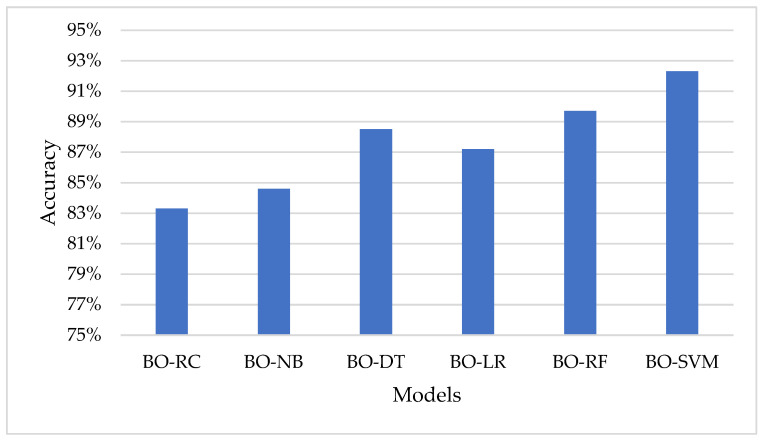
Representation of the models in the term of accuracy using the Bayesian Optimization approach.

**Figure 8 sensors-23-02085-f008:**
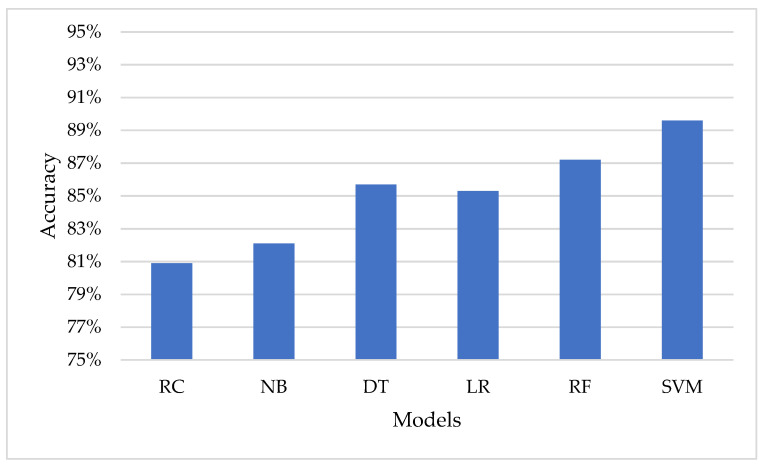
Representation of the models in the term of accuracy using the default parameters.

**Figure 9 sensors-23-02085-f009:**
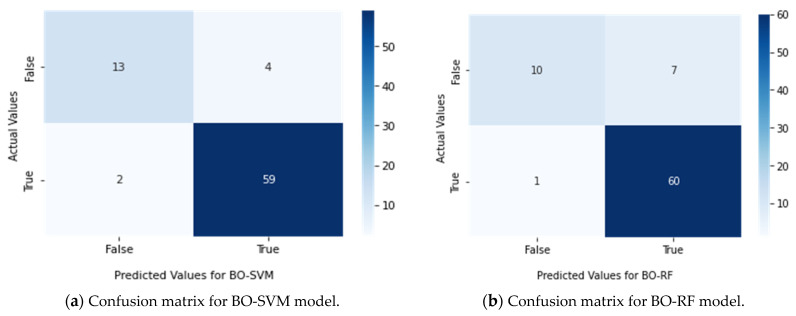
Confusion matrix for classifiers (**a**) BO-SVM, (**b**) BO-RF, (**c**) BO-LR, (**d**) BO-DT, (**e**) BO-NB, and (**f**) BO-RC.

**Figure 10 sensors-23-02085-f010:**
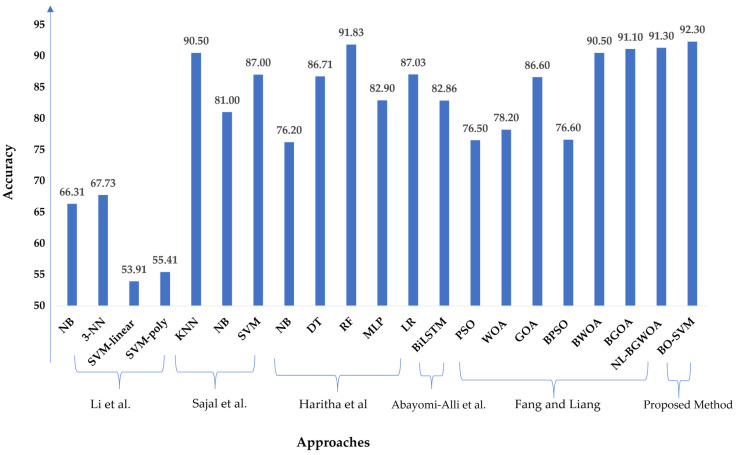
Comparison between the proposed model and the recent approaches [56,57,58,59,60] using the same standard PD dataset.

**Table 1 sensors-23-02085-t001:** Comparison between PD diagnosis techniques.

Ref	Objective	Data Source & Sample Size	Techniques	Outcome	Benefits	Limitations
[29]	Diagnosis and classification of PD from HC	Private dataset from 38 individuals 20 PD and 18 Health Control (HC)	C4.5 Decision Tree the extracted features using PCA	LOOCV = 63.20%	The fundamental issue in clinical practice is not so much distinguishing individuals with Parkinsonian disorders from healthy controls	A huge amount of patient data must be collected at various stages of illness development
[30]	Diagnosis and classification of PD from MCI	Private dataset 42 subjects from PD patients	Feature selection based on LASSO	MAE = 8.38	Because the new probability distributions span a wider normalized range, log-transformed measures outperformed their linear counterparts	Depending on linear regression which underperforms with non-linear decision boundaries
[31]	Diagnosis and classification of PD from HC	Private dataset from 10 medical centers of 10 PD patients	Deep Learning	R2 = 0.956	Using DL help in identifying more insights in the dataset	Small datasets affect model and robustness increase the training time
[32]	Diagnosis and classification of PD from MCI	Gait data from 15 IPD, 15 VaP, and 15 healthy participants were collected using wearable sensors placed on both feet	MLP DBN	ACC = 94.50% ACC = 93.50%	They utilized a classification approach based on two different classifiers MLP and DBN with a better performance such that the problem is the differentiation between VaP and IPD	In balance, dataset affects model generalization ability
[33]	Classification of patients with essential tremor (ET) from tremor-dominant Parkinson disease (PD)	13 PD patients (tremor dominant forms) and 11 ET patients	SVM	ACC of SVM with RBF = 83.00%	Classification of patients with essential tremor (ET) from tremor-dominant Parkinson disease (PD)	13 PD patients (tremor dominant forms) and 11 ET patients
[34]	Classification of PD from HC	UCI data include voice measurements of 31 people	DNN SVM And Fuzzy neural system	ACC = 81.03%	Use hybrid model between NN and fuzzy system	Very small dataset with trained samples
[35]	Classification of PD from HC	55 patients with Parkinson’s and 23 subjects with Parkinson’s related syndromes	(DNNs) with shared hidden layers.	ACC = 92.00%	Predicting Parkinson’s illness with multitask learning	Data on disease duration and treatment (i.e., treatment duration and dose)
[36]	Classification between PD and normal control	(56 PD, and 56 Normal Control)	Linear classifier	ACC = 89.00% F-Measure = 87.00%	Using iterative canonical correlation analysis for feature selection	The model needs to be tested with different train test splits
[37]	Classification between PD and HC	Handwriting of a sentence in 37 PD patients	SVM	ACC = −84.00%	Choosing optimal features based on sequential forward feature selection (SFFS)	Handwriting may overlap with other diseases

**Table 2 sensors-23-02085-t002:** Dataset features description.

Column Name	Description
Name/ASCII	Patient’s name/Record Num
MDVP-Fo (Hz)	Vocal fundamental (mean frequency)
MDVP Fhi (Hz)	Vocal fundamental (Max frequency)
MDVP Flo (Hz)	Vocal fundamental (Min frequency)
MDVP jitter (%)	Several measurements differ in fundamental frequency (i.e., RAP, MDVP, APQ, etc.)
MDVP Fhi (Hz)	Several measurements differ in amplitude (i.e., APQ5, MDVP: APQ, etc.)
NHR, HNR	The ratio of noise with regard to total components in voice
RPDE, D2	Nonlinear complexity measurements
DFA	Fractal scaling exponent
PPE, spread1, spread2	Three nonlinear methods for calculating fundamental frequency variation

**Table 3 sensors-23-02085-t003:** Statistical analysis for the dataset.

	Count	Mean	Std	Min	50%	Max
MDVP:Fo(Hz)	195.0	154.2286	41.390065	88.33300	148.7900	260.1050
MDVP:Fhi(Hz)	195.0	197.1049	91.491548	102.1450	175.8290	592.0300
MDVP:Flo(Hz)	195.0	116.3246	43.521413	65.47600	104.3150	239.1700
MDVP:Jitter(%)	195.0	0.006220	0.004848	0.001680	0.004940	0.033160
MDVP:Jitter(Abs)	195.0	0.000044	0.000035	0.000007	0.000030	0.000260
MDVP:RAP	195.0	0.003306	0.002968	0.000680	0.002500	0.021440
MDVP:PPQ	195.0	0.003446	0.002759	0.000920	0.002690	0.019580
Jitter:DDP	195.0	0.009920	0.008903	0.002040	0.007490	0.064330
MDVP:Shimmer	195.0	0.029709	0.018857	0.009540	0.022970	0.119080
MDVP:Shimmer(dB)	195.0	0.282251	0.194877	0.085000	0.221000	1.302000
Shimmer:APQ3	195.0	0.015664	0.010153	0.004550	0.012790	0.056470
Shimmer:APQ5	195.0	0.017878	0.012024	0.005700	0.013470	0.079400
MDVP:APQ	195.0	0.024081	0.016947	0.007190	0.018260	0.137780
Shimmer:DDA	195.0	0.046993	0.030459	0.013640	0.038360	0.169420
NHR	195.0	0.024847	0.040418	0.000650	0.011660	0.314820
HNR	195.0	21.88597	4.425764	8.441000	22.085000	33.04700
RPDE	195.0	0.498536	0.103942	0.256570	0.495954	0.685151
DFA	195.0	0.718099	0.055336	0.574282	0.722254	0.825288
spread1	195.0	−5.684397	1.090208	−7.964984	−5.720868	−2.434031
spread2	195.0	0.226510	0.083406	0.006274	0.218885	0.450493
D2	195.0	2.381826	0.382799	1.423287	2.361532	3.671155
PPE	195.0	0.206552	0.090119	0.044539	0.194052	0.527367
Status	195.0	0.753846	0.431878	0.000000	1.000000	1.000000

**Table 4 sensors-23-02085-t004:** Hyperparameters tuning for the classification models using the Bayesian Optimization approach.

Models	Best Hyperparameters
RF	N_estimators = 10, criterion = gini.
RC	Alpha = 0.4, copy_X = false, fit_intercept = true, normalize = false, solver = lsqr, tol = 0.01.
DT	Criterion = entropy, splitter = random.
NB	Alpha = 0.1, var_smoothing = 0.00001.
LR	Penalty = l2, solver = lbfgs.
SVM	Kernel = rbf, regularization parameter (C) = 0.4.

**Table 5 sensors-23-02085-t005:** Performance of the classification models using the Bayesian Optimization approach.

Models	Accuracy	F1 Score	Recall	Precision
BO-RC	83.3%	82.2%	83.3%	82.0%
BO-NB	84.6%	84.4%	86.6%	84.5%
BO-DT	88.5%	87.7%	88.5%	88.0%
BO-LR	87.2%	86.5%	87.2%	86.5%
BO-RF	89.7%	88.9%	89.7%	89.8%
BO-SVM	92.3%	92.1%	92.3%	92.1%

**Table 6 sensors-23-02085-t006:** Performance of the classification models in the term of accuracy using the default parameters.

Models	Accuracy
RC	80.9%
NB	82.1%
DT	85.7%
LR	85.3%
RF	87.2%
SVM	89.6%

## Data Availability

This dataset is taken from publicly available database [45].

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
