# Peer review of "Bayesian Optimization with Support Vector Machine Model for Parkinson Disease Classification"

_sensors, 2023, doi:10.3390/s23042085_

Round 1

Reviewer 1 Report

The paper presents a new approach to classify people suffering from Parkinson's disease (PD) and those who do not, by using Bayesian Optimization-Support Vector Machine (BO-SVM). The approach involves the use of Bayesian Optimization (BO), a hyperparameter tuning technique, to optimize the hyperparameters for six different machine learning models, namely, Support Vector Machine (SVM), Random Forest (RF), Logistic Regression (LR), Naive Bayes (NB), Ridge Classifier (RC), and Decision Tree (DT). The dataset used in this study consisted of 23 features and 195 instances, and the performance of the models was evaluated using four metrics: accuracy, F1-score, recall, and precision.

The results showed that SVM model achieved the best performance among all the models, both before and after the hyperparameter tuning process, with an accuracy of 92.3% obtained using BO. The study can be considered as a useful contribution to the field of machine learning and its applications in healthcare.

However, it is also important to note that while the results of this study are promising, they need to be further validated on a larger dataset and by considering other evaluation metrics. Additionally, the study did not address how the results could be generalized to a larger population or how the proposed approach could be integrated into a larger healthcare system.

The overview of related works must be improved by discussing more recent studies such as doi:10.1109/ACCESS.2020.2995737. Explain what we can see in the heatmap (Figure 3). Figure 4 is run away. Add a discussion section to discuss the limitations of the proposed methodology.

Overall, the study provides a foundation for future research in the field of machine learning for healthcare, and the BO-SVM approach shows potential as a useful tool for diagnosing PD. However, further research is necessary to validate the results and to assess the generalizability of the approach.

Reviewer 2 Report

There are several hyperparameter optimization (HPO) tools. We are using Optuna, SigOpt, NNI etc. Why have you chosen BO? Whether BO carries any distinct advantages when compared with other HPO methods. Will those ML algorithms give better results when optimized other HP methods? What is the source of the dataset? 

Round 2

Reviewer 1 Report

The authors addressed a majority of my comments and improved the article accordingly. Few issues, however remain:

- Explain what kind of cross-validation was used. Why you did not use LOOCV?

- More explanation should be given on what we can see in Figures 3, 5 and 6. Any specifics what the reader should take attention? More in-depth analysis and discussion is needed.

- Figure 8: a line plot should not be used when an independent variable is categorical. Replace with a bar plot.

- Only two studies by other authors were used for comparison in Table 10. The UCI dataset however is widely used and many studies have analyzed it. The authors should include for comparison more studies, for example, DOI 10.1007/s42235-022-00253-6, DOI: 10.15439/2020F188.

Reviewer 2 Report

The authors have responded positively and promptly to all the suggestions and corrections recommended by all the reviewers. My question of why you have gone for BO as the hyperparameter optimization (HPO) tuning tool is not yet answered in the revised version of the paper. There are many HPO tools in the industry and why the authors specifically focus on BO needs a solid explanation

Round 3

Reviewer 1 Report

The authors have revised well. The manuscript can be accepted for publication.